# Force generation by a propagating wave of supramolecular nanofibers

Ryou Kubota [1], Masahiro Makuta[2], Ryo Suzuki [3], Masatoshi Ichikawa [2], Motomu Tanaka [3,4] & Itaru Hamachi [1,5✉]

Dynamic spatiotemporal patterns that arise from out-of-equilibrium biochemical reactions generate forces in living cells. Despite considerable recent efforts, rational design of spatiotemporal patterns in artificial molecular systems remains at an early stage of development. Here, we describe force generation by a propagating wave of supramolecular nanofibers. Inspired by actin dynamics, a reaction network is designed to control the formation and degradation of nanofibers by two chemically orthogonal stimuli. Real-time fluorescent imaging successfully visualizes the propagating wave based on spatiotemporally coupled generation and collapse of nanofibers. Numerical simulation indicates that the concentration gradient of degradation stimulus and the smaller diffusion coefficient of the nanofiber are critical for wave emergence. Moreover, the force (0.005 pN) generated by chemophoresis and/or depletion force of this propagating wave can move nanobeads along the wave direction.

[1] Department of Synthetic Chemistry and Biological Chemistry, Graduate School of Engineering, Kyoto University, Katsura, Nishikyo-ku, Kyoto 615-8510, Japan. [2] Department of Physics, Kyoto University, Sakyo-ku, Kyoto 606-8502, Japan. [3] Center for Integrative Medicine and Physics, Institute for Advanced Study, Kyoto University, Sakyo-ku, Kyoto 606-8501, Japan. [4] Physical Chemistry of Biosystems, Institute of Physical Chemistry, Heidelberg University, 69120 Heidelberg, Germany. [5] JST-ERATO, Hamachi Innovative Molecular Technology for Neuroscience, Kyoto University, Katsura, Nishikyo-ku, Kyoto 615-8530, Japan. ✉email: ihamachi@sbchem.kyoto-u.ac.jp

Out-of-equilibrium events in chemistry and biology are abundant in nature at all scales, providing a variety of spatial-temporal patterns, such as oscillations, waves, and spirals[1–4]. In living cells, dynamic spatiotemporal patterns play essential roles in cellular functions, such as Min systems for division and actin waves for migration[5–7]. Force is also produced by such spatiotemporal patterns[5,6]. In lamellipodia, the propagating wave of actin filaments can generate the force required to push the cell membrane forward for cell migration, as has been well studied by single-molecule force spectroscopy[8–11].

Inspired by these cellular systems, out-of-equilibrium dynamic patterns of supramolecular architectures are anticipated to give life-like characteristics, such as autonomy, adaptivity, and homeostasis, to artificial soft materials[12–38]. For example, polymer-based soft materials coupled with the Belousov–Zhabotinsky reaction can exhibit sophisticated functions such as self-oscillation of gels, self-walking actuators, and autonomous mass transport[39–42]. Recently, formation of spatiotemporal patterns relying on stimulus-responsive supramolecular nanofibers has been proposed. A fuel-driven dissipative supramolecular system, pioneered by van Esch et al., showed stochastic formation and degradation of supramolecular nanofibers, albeit not coupled in time and space[22]. Hermans et al. reported a traveling front of supramolecular colloids in a non-stirred reaction vessel; however, the degradation did not occur simultaneously[31]. Despite considerable efforts, rational guidelines have yet to be determined for generating spatiotemporal patterns based on formation and degradation of supramolecular nanofibers. Moreover, precise measurements of the forces generated by artificial spatiotemporal patterns have yet to be achieved. There have been many examples of force generation by out-of-equilibrium systems in biology; however, it is reasonable to expect that artificial spatiotemporal patterns of supramolecular nanofibers can also generate forces. Such efforts might bridge biological and artificial systems, which is a crucial step towards design of next-generation soft materials with cell-mimetic functions.

We herein describe force generation by a propagating wave of supramolecular nanofibers comprising a peptide-type hydrogelator (Fig. 1a). In lamellipodia of cell migration, distinct accessory proteins such as Arp2/3 complex and ADF/cofilin independently control the kinetics of actin polymerization and depolymerization in a spatiotemporal manner[5,6]. Inspired by actin dynamics, we design a reaction network to control the formation and degradation of supramolecular nanofibers comprising a short peptide by two chemically orthogonal stimuli. Real-time confocal laser scanning microscopic (CLSM) imaging visualizes that the propagating wave based on spatiotemporally coupled generation and collapse of nanofibers proceeds in the mm scale. Numerical simulation and analysis indicate that the concentration gradients of the formation and degradation stimuli are one of the main controlling factors and the smaller diffusion coefficient of the nanofiber is critical for wave emergence. Moreover, we succeed in quantitative determination of the force generated by this propagating wave, which can move nanobeads along the wave direction.

## Results

**Design of a chemical reaction network for a propagating wave.** To realize the supramolecular propagation wave by imitating dynamic actin assembly/disassembly, we set the following four guidelines to design a chemical reaction network for an artificial propagating wave. First, the propagating wave is driven by a concentration gradient of chemical stimuli. Second, two distinct chemical stimuli can induce formation or degradation of the nanofibers. Third, a monomer has two different functional groups, each of which orthogonally responds to one distinct stimulus.

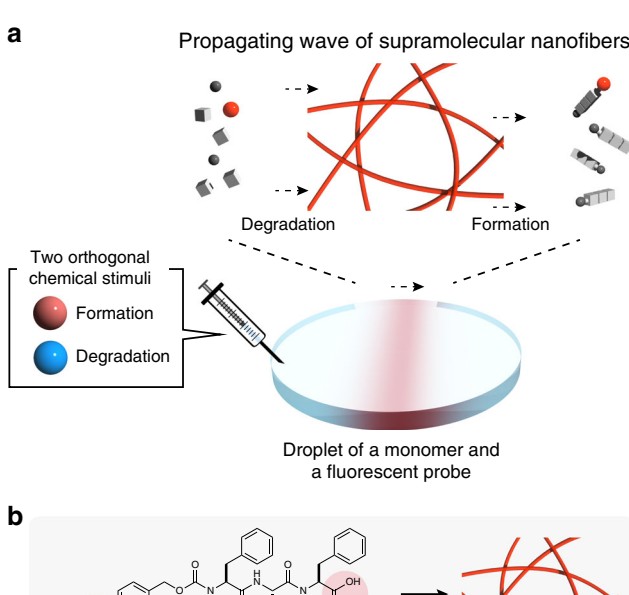

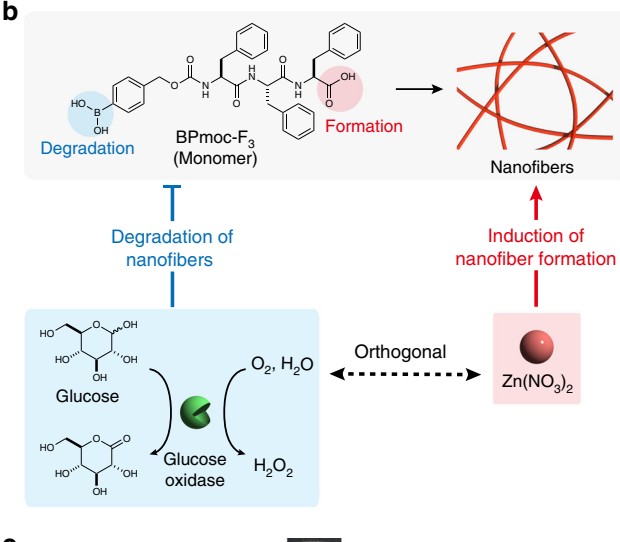

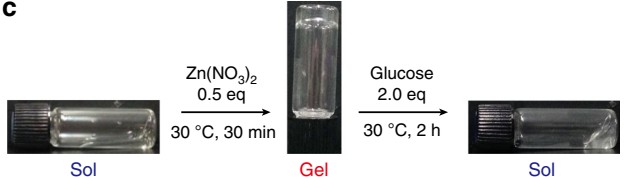

**Fig. 1 Design of a propagating wave of supramolecular nanofibers.**
**a** Schematic illustration of a propagating wave comprising spatiotemporally coupled formation and degradation of supramolecular nanofibers driven by the concentration gradient of two chemically orthogonal stimuli. **b** A reaction network for the supramolecular propagating wave. $Zn(NO_3)_2$ induces the nanofiber formation, while $H_2O_2$ produced by enzymatic reaction of glucose oxidase with glucose collapses the nanofibers. The formation and degradation stimuli are chemically orthogonal to each other. **c** Macroscopic sol-gel-sol transition of BPmoc-F$_3$ containing glucose oxidase (GOx) upon addition of $Zn(NO_3)_2$ (0.5 eq) and glucose (2.0 eq). Condition: [BPmoc-F$_3$] = 2.4 mM, [GOx] = 1 mg/mL, [$Zn(NO_3)_2$] = 1.2 mM, [glucose] = 4.8 mM, 50 mM HEPES, pH 7.4.

Fourth, the two stimuli should not interfere with each other to enable precise control of the reaction conditions. Here, we selected BPmoc-F$_3$ hydrogelator as a monomer (Fig. 1b). BPmoc-F$_3$ has two functional groups for formation and degradation of supramolecular nanofibers: carboxylate and boronobenzyl groups at the C- and N-termini, respectively. BPmoc-F$_3$ forms supramolecular nanofibers in an aqueous buffer solution, and the BPmoc-F$_3$ nanofibers decompose upon treatment of oxidase/substrate pairs (e.g., glucose oxidase (GOx) and glucose) through a 1,6-elimination reaction on the

boronobenzyl oxycarbonyl group by $H_2O_2$ (Fig. 1b, Supplementary Fig. 1)[43,44]. For the formation stimulus we used $Zn^{2+}$ ions because the carboxylate group acts as a coordination site to trigger nanofiber formation by bridging BPmoc-$F_3$ monomers and/or suppressing anionic charge repulsion. Furthermore, the $Zn^{2+}$ ion is redox-inert so that it will not interfere with glucose/GOx stimulus (generating $H_2O_2$)[45]. Therefore, it is conceivable that the $Zn^{2+}$ ion and GOx/glucose are orthogonal stimuli that might induce formation and decomposition of BPmoc-$F_3$ nanofibers, respectively.

**Stimulus-responsive nanofiber formation and degradation.** We initially investigated $Zn^{2+}$ ion-triggered hydrogelation of BPmoc-$F_3$ by a tube inversion method. After addition of 0.5 eq of $Zn^{2+}$ ion, an aqueous HEPES buffer solution of BPmoc-$F_3$ changed into a transparent hydrogel within 30 min (Fig. 1c, Supplementary Fig. 2a). The resulting hydrogel turned back to a sol state upon addition of 0.5 eq of EDTA. We also confirmed that neither a sol–gel nor a gel–sol transition took place upon addition of $H_2O$ or buffer instead of $Zn(NO_3)_2$ or EDTA solution, respectively (Supplementary Fig. 2b, c). These results indicated the potential for coordination-driven hydrogelation of BPmoc-$F_3$. We next examined the responsiveness of the $Zn^{2+}$-induced BPmoc-$F_3$ nanofibers towards a pair of GOx and glucose. The $Zn^{2+}$-induced hydrogel containing GOx changed into the solution state after incubation at 30 °C for 2 h in the presence of glucose (2.0 eq) (Fig. 1c). As a control, the hydrogel maintained the gel state upon addition of the same amount of $H_2O$ (Supplementary Fig. 3a). HPLC analyses after glucose treatment demonstrated that BPmoc-$F_3$ was completely decomposed within 2 h, whereas over 97% of BPmoc-$F_3$ remained upon $H_2O$ treatment (Supplementary Fig. 3b, c).

We then performed real-time CLSM imaging of the formation and degradation of $Zn^{2+}$-induced BPmoc-$F_3$ nanofibers stained with BP-TMR, a fluorescent probe (Supplementary Fig. 4). After addition of $Zn(NO_3)_2$ (0.5 eq) to a homogeneous solution of BPmoc-$F_3$ and BP-TMR in HEPES buffer, thin fibrous nanofibers gradually formed (Supplementary Fig. 5a, Supplementary Movie 1). Observation of the formation process showed that aggregate-like seeds stochastically formed, and then nanofibers grew from the seeds to connect the different seeds/fibers, resulting in the formation of a fiber network (Supplementary Fig. 5b). The fluorescence intensity profile of the CLSM images revealed that the nanofiber formation continued for over 60 min (Supplementary Fig. 5c). Furthermore, real-time CLSM imaging allowed us to visualize the fiber degradation process upon addition of glucose (2.0 eq) to the hydrogel containing GOx (Supplementary Figs. 6a and 7, Supplementary Movie 2). A time-course plot of the fluorescent intensity changes showed that $Zn^{2+}$-induced nanofibers homogeneously disappeared over 60 min (Supplementary Fig. 6c). We also confirmed that the interaction between $Zn(NO_3)_2$ and the GOx/glucose pair was negligible in nanofiber formation and degradation processes (Supplementary Figs. 8 and 9).

**Real-time imaging of a supramolecular propagating wave.** With formation and degradation stimuli of the supramolecular nanofibers in hand, we attempted to generate the propagating wave of supramolecular nanofibers upon simultaneous treatment of $Zn(NO_3)_2$ and glucose. To observe the propagating wave by real-time CLSM imaging, we prepared a droplet containing BPmoc-$F_3$, BP-TMR, and GOx, which was sandwiched between two glass plates (Fig. 2a). Time-lapse CLSM imaging after addition of a mixture of $Zn(NO_3)_2$ (0.5 eq) and glucose (2.0 eq) via a syringe at the right edge of the droplet clearly visualized the generation of the propagating wave based on spatially-coupled formation and degradation of supramolecular nanofibers (Fig. 2b,

Supplementary Movie 3). Immediately after addition of $Zn(NO_3)_2$ and glucose, small seeds stochastically formed. Thereafter (around 10 min), the nanofibers gradually appeared from the right side and grew up towards the left side, followed by degradation of the supramolecular nanofibers along the same direction. After the wave propagation, no notable fluorescence was observed, suggesting that the BPmoc-$F_3$ fibers were completely decomposed. The time course of the fluorescent intensity at distinct $x$ coordinates quantitatively demonstrated that the time of the maximum fluorescent intensity was delayed as the $x$-coordinate increased (Fig. 2c). Hence, the supramolecular wave clearly propagated along the $x$ axis. In contrast to simultaneous growth and shrinkage of nanofibers in proximity reported by van Esch[22], nanofiber formation and degradation in this propagating wave were highly regulated in a spatiotemporal manner by the concentration gradient of stimuli, which would suppress the stochastic nature of the fiber formation/degradation process.

To investigate the behavior of the propagating wave in detail, we next conducted CLSM imaging over a wider field of view (Fig. 2e, Supplementary Movie 4). After addition of a mixture of $Zn(NO_3)_2$ ions and glucose, a crescent-shaped bright area immediately appeared. After 15 min, the fluorescence at the right edge of the crescent region became brighter. The increase of the fluorescent intensity propagated to the left, whereas the fluorescent intensity at the original position decreased. To quantitatively assess the behavior of the supramolecular propagating wave, we plotted the time course of the fluorescent intensity changes at various distances from the original (Fig. 2f). As a typical example, the fluorescent intensity at $x = 1.3$ mm gradually increased during the initial 5 min, likely because of nanofiber formation (Fig. 2f, red line). After 25 min, the fluorescent intensity increased nonlinearly and reached its maximum at 30 min. Then, the fluorescent intensity decreased for the next 5 min. The time course plot reveals that a propagating wave was generated in the macroscopic range (mm scale). The wave started at $x = 1.1$ mm and completely disappeared at $x = 1.5$ mm. On the basis of these observations, the propagating distance and the average velocity of the propagating wave were estimated to be $340 \pm 40$ μm and $54 \pm 8$ μm/min, respectively (Supplementary Fig. 10).

The formation of the supramolecular propagating wave was strongly dependent on the reaction conditions. When half concentration of GOx was used (0.5 mg/mL), the propagating wave was indeed formed, while the duration of the propagating wave slightly increased and the velocity decreased to be $20 \pm 2$ μm/min (Supplementary Fig. 11, Supplementary Movie 5). On the other hand, a propagating wave was not observed when a smaller amount of glucose (1.0 eq) was used (Supplementary Fig. 12a, Supplementary Movie 6). CLSM imaging showed that the nanofibers were indeed formed over 60 min immediately after addition of the two stimuli; however, the system degraded homogenously. The time course change of the fluorescent intensity was independent of the $x$-coordinates, suggesting that the propagating wave did not form under these conditions (Supplementary Fig. 12b). When a larger amount of GOx (2 mg/mL) or glucose (63 eq) was used, we observed negligible formation of nanofibers owing to rapid degradation of BPmoc-$F_3$ (Supplementary Figs. 13 and 14). To confirm whether simultaneous addition of $Zn(NO_3)_2$ and glucose is important for the wave formation, we treated the droplets of BPmoc-$F_3$, BP-TMR, and GOx by addition of glucose 30 min after $Zn(NO_3)_2$ injection (nanofiber formation is saturated after incubation for 30 min as shown in Supplementary Fig. 5). The BPmoc-$F_3$ nanofibers formed upon treatment with $Zn(NO_3)_2$ and begun to homogenously disappear 50 min after addition of glucose, indicating no generation of a propagating wave (Fig. 2d, Supplementary Fig. 15, Supplementary Movie 7).

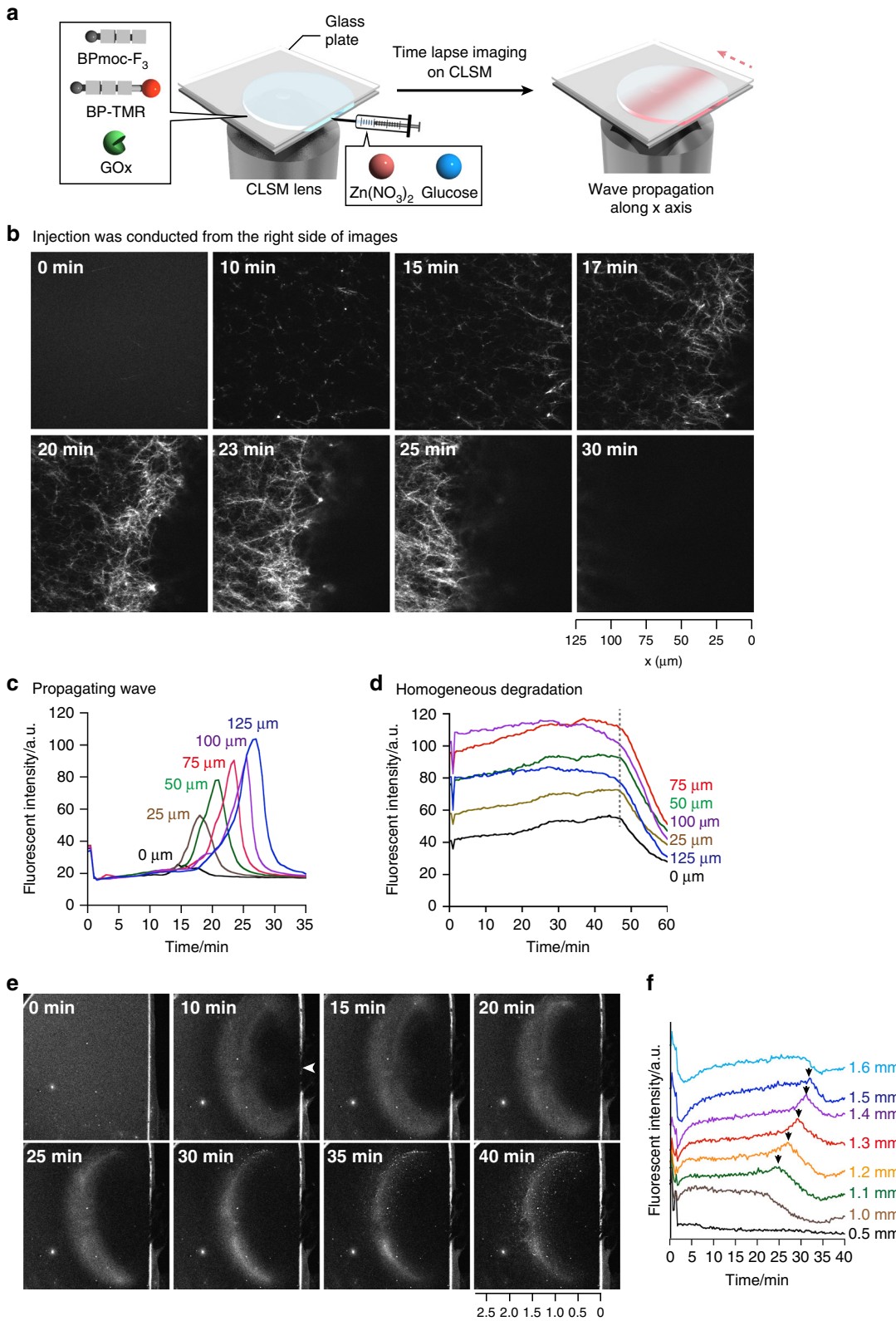

It implies that concentration gradient of Zn(NO$_3$)$_2$ would play an important role for wave formation. We also confirmed that pretreatment of the glucose before addition of Zn$^{2+}$ ions did not induce nanofiber formation (Supplementary Fig. 16). These observations indicated that simultaneous addition of Zn$^{2+}$ and glucose at a critical concentration would kinetically balance the formation/degradation of the supramolecular fibers, which is crucial for the generation of the propagating wave.

**Numerical simulation of the supramolecular propagating wave.** To gain further insights into the supramolecular propagating

**Fig. 2 In situ real-time imaging of a propagating wave of supramolecular nanofibers. a** Schematic illustration of the experimental setup. A solution of BPmoc-F$_3$, BP-TMR, and GOx was placed between two glass plates. One minute after starting confocal laser scanning microscopic (CLSM) observation, two stimuli were added via a syringe from the right edge of the droplet. **b, e** Time-lapse CLSM imaging of the propagating wave of supramolecular nanofibers observed by **b** 100× and **e** 4× objectives. **b** A solution of Zn(NO$_3$)$_2$ and glucose was injected from the right side of images, and **e** the injection point was highlighted by an arrowhead. **c, d, f** Time-course of fluorescent intensity at distinct $x$-coordinates under the propagating wave (**c** Fig. 2b and **f** 2e) and **d** homogeneous nanofiber degradation (Supplementary Fig. 15). The shape of the crescent-shaped area varied depending on experiments because injection of the chemical stimuli was conducted manually; nevertheless the formation of the propagating wave is highly reproducible (Supplementary Fig. 10). Condition: [BPmoc-F$_3$] = 1.6 mM, [BP-TMR] = 0.34 μM, [GOx] = 1.0 mg/mL, [Zn(NO$_3$)$_2$] = 0.8 mM, [glucose] = 3.2 mM in 50 mM HEPES, pH 7.4, 30 °C.

wave, we performed a numerical simulation based on a reaction-diffusion model. Because of the formation processes of the supramolecular nanofibers, we supposed three reaction steps: (i) Zn$^{2+}$-promoted nanofiber formation between monomers, (ii) Zn$^{2+}$-dependent and (iii) Zn$^{2+}$-independent nanofiber elongation processes. These formation processes are linearized. The decomposition of the monomer and nanofibers were assumed to be a second-order reaction based on the degradation stimulus. The interference between the formation and degradation stimuli was ignored owing to their chemical orthogonality. The concentrations of the stimuli were decreased only by molecular diffusion because the formation stimulus should be reused and an excess amount of degradation stimulus was added. From these assumptions, we obtained a set of reaction-diffusion equations:

$$\frac{\partial n}{\partial t} = D_n \nabla^2 n + k_1 mx + k_2 nm - k_3 ny + k_5 nmx \tag{1}$$

$$\frac{\partial m}{\partial t} = D_m \nabla^2 m - k_1 mx - k_2 nm - k_4 my \tag{2}$$

$$\frac{\partial x}{\partial t} = D_x \nabla^2 x \tag{3}$$

$$\frac{\partial y}{\partial t} = D_y \nabla^2 y \tag{4}$$

where, $n$, $m$, $x$, and $y$ are defined as concentrations of the nanofibers, monomer, formation, and degradation stimuli, respectively. For the numerical simulation, we considered that the $k_2$ value and the diffusion coefficient of the nanofibers should be much smaller than other kinetic and diffusion coefficients. Based on these equations, we successfully reproduced the propagating wave under the appropriate conditions, as shown in Fig. 3a and Supplementary Movie 8. The numerical simulation demonstrates that a band of supramolecular nanofibers with limited width proceeded to the left, as shown in the kymograph (Fig. 3b). In addition, the monomer had a spatial concentration gradient, which strongly depended on the concentration gradient of the formation and degradation stimuli. This result suggests that these concentration gradients were one of the main driving forces for wave formation. Numerical analysis also provided other important insights into the propagating wave; the wave velocity was proportional to the formation kinetics of the nanofibers and the diffusion coefficient of the degradation stimulus, and the initial anisotropic shapes of the formation and degradation stimuli rapidly decayed into half-round distributions and were not crucial to the crescent-shaped propagation. In addition, it is implied that the concentration gradient of the degradation stimulus is a key parameter for maintaining wave propagation (see SI for details). Furthermore, the much smaller diffusion coefficient of the nanofibers is another important factor for producing a propagating wave, suggesting that a supramolecular nanofiber is one of the most suitable scaffolds for spatiotemporal pattern formation. The numerical simulation also suggested that the small

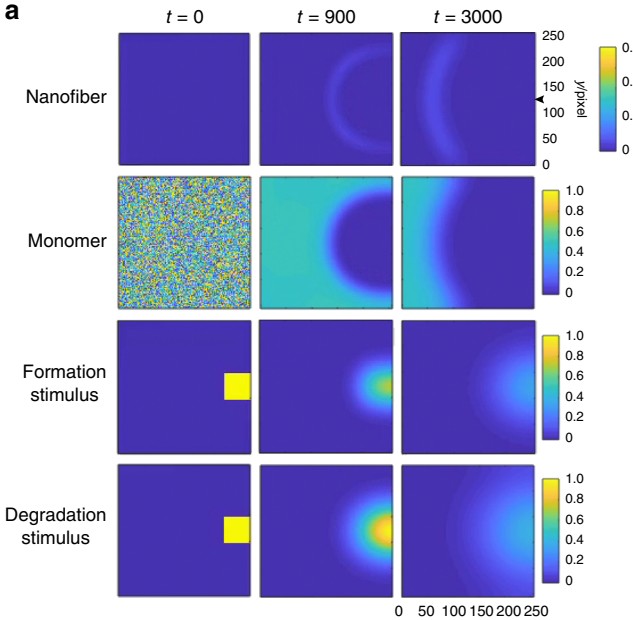

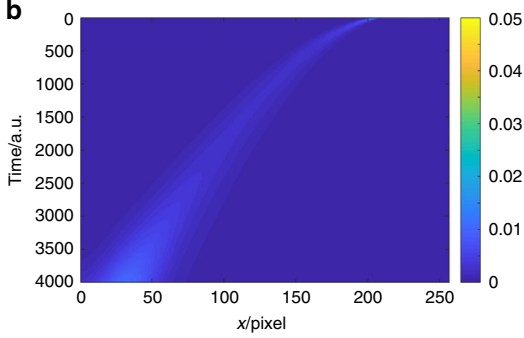

**Fig. 3 Numerical simulation of the propagating wave based on a reaction-diffusion mechanism. a** Time dependent concentration changes of (top) supramolecular nanofibers, (2nd row) monomer, (3rd row) formation, and (bottom) degradation stimuli. Please see the main text and SI for the reaction-diffusion equations. **b** Kymograph of the nanofiber concentration on the center at $y$ axis ($y = 128$, highlighted by a black arrowhead) of Fig. 3a. Condition: $D_n$: 0.0001, $D_m$: 0.003, $D_x$: 0.003, $D_y$: 0.006, $k_1$: 0.1, $k_2$: 0.01, $k_3$: 0.4, $k_4$: 0.2, $k_5$: 0.1.

perturbation of diffusion (by microfluidics, for example) would enable the modulation of wave propagation. Our numerical simulation demonstrates that the design principle of the reaction network based on orthogonal formation and degradation stimuli is a promising strategy for forming supramolecular propagating waves.

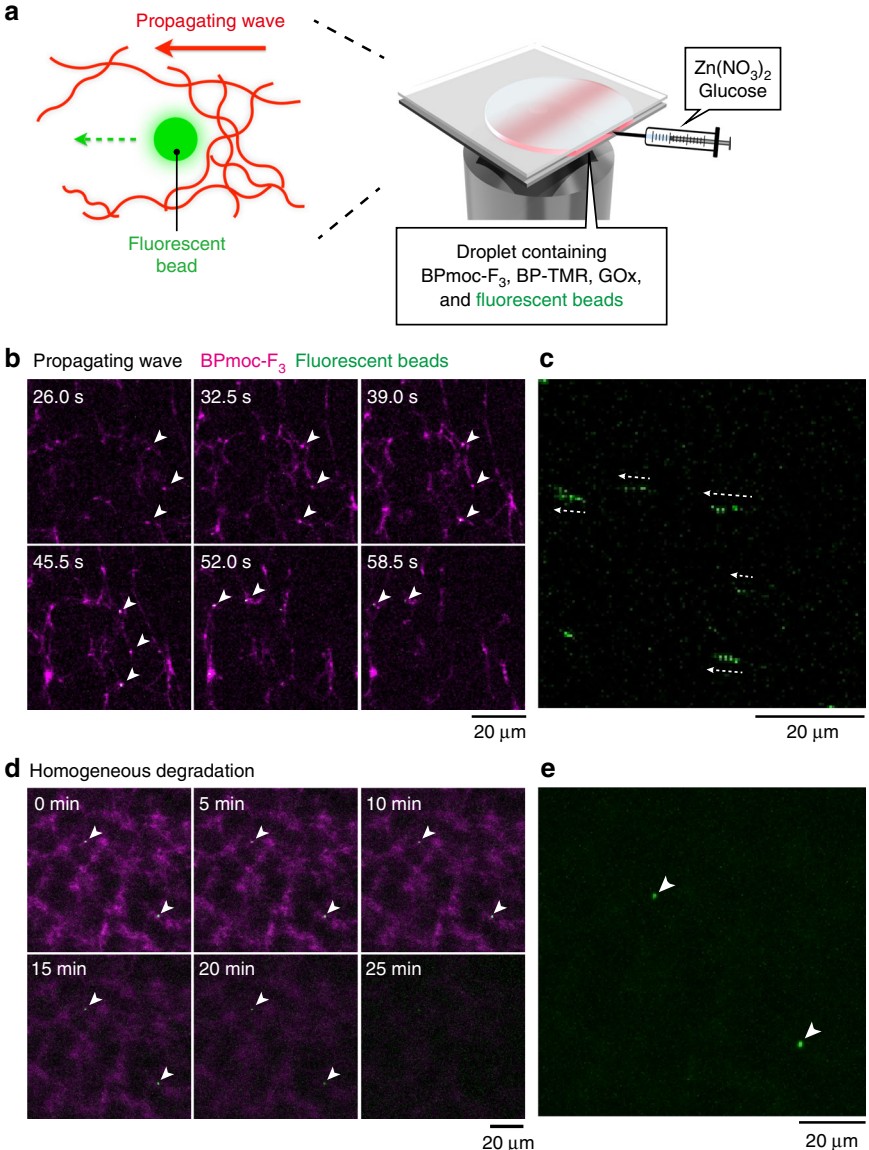

**Fig. 4 Force generation by the supramolecular propagating wave. a** Schematic illustration of the bead movement induced by the propagating wave. **b**, **d** Time-lapse images of the bead displacement under **b** the propagating wave and **d** homogeneous degradation of supramolecular nanofibers. Time at **b** starting the time-lapse imaging or **d** adding glucose was defined as 0. **c**, **e** Trajectories of the intensity maxima from the fluorescently labeled beads under **c** the propagating wave and **e** homogeneous nanofiber degradation monitored over **c** 26 s and **e** 25 min, respectively. White arrowheads show the position of fluorescently labeled beads. Green: fluorescently labeled beads, magenta: BPmoc-F$_3$/BP-TMR. As shown in Supplementary Fig. 17, time course of fluorescent intensity of nanofibers showed that the propagating wave formed in Fig. 4b. Time-lapse CLSM images of the bead displacement under homogeneous nanofiber formation were shown in Supplementary Fig. 18. Condition: [BPmoc-F$_3$] = 1.6 mM, [BP-TMR] = 0.34 μM, [GOx] = 1.0 mg/mL, [Zn(NO$_3$)$_2$] = 0.8 mM, [beads] = 20 μg/mL, [glucose] = 3.2 mM in 50 mM HEPES, pH 7.4, 30 °C.

**Force generation by the supramolecular propagating wave**. We finally attempted to examine if a force is generated by this supramolecular propagating wave. The experiment and simulation predicted that the propagating wave should generate the forces that depends on the gradient of chemical (chemophoresis), and that depends on the concentration gradient of the fibrous supramolecules which depletes larger objects from the higher concentrated region (depletion or entropy effects). To evaluate the force quantitatively, we monitored the displacement of PEG-coated fluorescently labeled polystyrene beads (diameter: 500 nm) during the wave propagation (Fig. 4a). Real-time CLSM imaging clearly indicated that the beads move along the direction of the propagating wave. As shown in Fig. 4b, c, and Supplementary Movie 9, multiple beads appeared in the field of view at the

beginning of the wave formation and continued to move during the propagation, resulting in a displacement of *ca*. 10 μm. The time-profile of the trajectory revealed the average velocity of the beads was *ca*. 0.4 μm/s. Taking the viscosity of the medium to be *ca*. 2.5 mPa·s, the minimal force generated by the propagation wave was calculated by the Stokes law to be *ca*. 0.005 pN (see *Methods* for the detailed calculation). As a control, we confirmed that the homogeneous nanofiber formation and degradation did not cause notable movement of the beads (Fig. 4d, e, Supplementary Fig. 18, Supplementary Movies 10 and 11). To the best of our knowledge, this is the first example of quantitative determination of the force generated by the spatiotemporal pattern of artificial supramolecular systems. Compared with the biological examples [the stalling force of actin and microtubule

polymerization (0.76 and 2.7 pN per single filament, respectively)], the force generated by our supramolecular propagating wave is rather small[9,11]. To examine if our fibers are soft and undergo bending deformation, we measured the persistence length of the supramolecular nanofiber based on thermal fluctuation (Supplementary Fig. 19, Supplementary Movie 12)[46]. The persistence length of our nanofibers ($\xi_{fiber} \approx 25 \, \mu m$) is comparable to that of actin filaments ($\xi_{actin} \approx 18 \, \mu m$)[47]. Because our current supramolecular wave consists of physically entangled nanofibers, more precise (vectorial) control of the supramolecular nanofiber structures (e.g., crosslinked, bundled, and branched networks[48]) will enable us to cover a range of forces comparable to those in biological systems (pN to nN order)[10]. Such behaviors have not been realized by dissipative synthetic assembly to date.

## Methods

**General**. Unless stated otherwise, all commercial reagents were used as received. Thin-layer chromatography (TLC) was performed on silica gel 60F$_{254}$ (Merck). $^1$H NMR spectra were obtained on a Varian Mercury 400 spectrometer with residual non-deuterated solvents (CD$_3$OD: 3.31 ppm for $^1$H) as the internal reference. ESI mass spectra were recorded using an Exactive (Thermo Scientific). Reversed-phase HPLC (RP-HPLC) was carried out on a Hitachi Chromaster system equipped with a diode array and YMC-Pack Triart C18 or ODS-A columns. All runs used linear gradients of acetonitrile (ACN) containing 0.1% trifluoroacetic acid (TFA) and 0.1% aqueous TFA. The images of confocal laser scanning microscopy (CLSM) were acquired by a FV1000 (Olympus) and a LSM800 (Carl Zeiss Microscopy). UPlanSApo 100× (1.40 numerical aperture, oil immersion, Olympus), UPlanSApo 4× (0.16 numerical aperture, Olympus), Plan-Apochromat 20× (0.8 numerical aperture, Carl Zeiss), and αPlan-Apochromat 100× (1.46 numerical aperture, Carl Zeiss) were used. The fluorescent intensity of the CLSM images was calculated by Fiji[49].

**Preparation of a homogeneous BPmoc-F₃ solution**. A BPmoc-F₃ powder was suspended in an aqueous buffer (50 mM HEPES, pH 7.4), and heated until dissolving. After cooling to room temperature, 10× stock solutions of BP-TMR (3.4 μM in 50 mM HEPES, pH 7.4) and/or GOx (10 mg/mL in 50 mM HEPES, pH 7.4 containing 5% (v/v) glycerol) were added.

**Zn²⁺ ion-induced gelation of BPmoc-F₃**. To a freshly-prepared aqueous solution of BPmoc-F₃ (2.4 mM, 0.15 wt%, 50 mM HEPES buffer (pH 7.4), 100 μL) was added a 10× stock solution of Zn(NO₃)₂ (12 mM in H₂O, 10 μL) or H₂O (10 μL). The resulting solution was allowed to stand for 30 min at room temperature. After confirming gelation, a 10× stock solution of EDTA (12 mM, 50 mM HEPES, pH 7.4, 10 μL) or buffer (10 μL) was added to the hydrogel, and incubated for 30 min at room temperature. The sample state (gel or sol) was confirmed by the tube inversion method. Photos were taken by iPhone 5 (Apple Inc.).

**Glucose response of GOx-encapsulated Zn²⁺-induced hydrogels**. To a freshly-prepared aqueous solution of BPmoc-F₃ and GOx (2.4 mM and 1 mg/mL, respectively, 50 mM HEPES containing 0.5% glycerol, pH 7.4, 100 μL) was added a 10× stock solution of Zn(NO₃)₂ (12 mM, 0.5 eq, 10 μL in H₂O). The resulting solution was incubated for 30 min at room temperature to form a transparent hydrogel. To the resultant hydrogel was added a 10× stock solution of glucose (48 mM, 2.0 eq, 10 μL in H₂O), and then the resulting hydrogel was incubated at 30 °C for 2 h. The state (gel or sol) was determined by the tube inversion method. To determine the reaction rate by RP-HPLC, a solution of p-nitrobenzenesulfonamide (1.6 mM, 100 μL in CH₃CN) was added to the gel/sol, and the resultant solution was analyzed by RP-HPLC.

**Real-time imaging of Zn²⁺-induced nanofiber formation**. A freshly-prepared aqueous solution of BPmoc-F₃ and BP-TMR (1.6 mM and 0.32 μM, respectively, 50 mM HEPES, pH 7.4, 10 μL) was deposited on a Matsunami glass-bottom dish (non-coated, thickness: 0.16~0.19 mm, catalog number: D11530H). A 10× stock solution of Zn(NO₃)₂ (8 mM, 1.0 μL in H₂O) or H₂O (1.0 μL) was added to the solution 1 min after starting time-lapse CLSM imaging. In Supplementary Fig. 8, a 10× stock solution of glucose (32 mM, 1 μL), GOx (10 mg/mL, 1 μL), or H₂O (1 μL) was added 30 min before Zn(NO₃)₂ addition. The images and time-lapse movies of the Zn²⁺-induced formation of supramolecular nanofibers were obtained by LSM800 with the 100× objective. The overall fluorescent intensity of the field of view was calculated by Fiji.

**Real-time imaging of glucose-responsive fiber degradation**. A freshly-prepared aqueous solution of BPmoc-F₃, BP-TMR, and GOx (1.6 mM, 0.32 μM and 1.0 mg/mL, respectively, 50 mM HEPES containing 0.5% glycerol, pH 7.4, 11 μL) was deposited on a Matsunami glass-bottom dish. To this solution was added a 10×

stock solution of Zn(NO₃)₂ (8 mM, 1.0 μL in H₂O). The resulting solution was incubated for 30 min at room temperature. A 10× stock solution of glucose (32 mM, 2.0 eq, 1.0 μL in H₂O) or H₂O was added one minute after starting the time-lapse imaging. The time-lapse movies of nanofiber degradation were obtained by FV1000 with the 100× objective. The overall fluorescent intensity of the field of view was calculated by Fiji.

**HPLC analysis of degradation kinetics of BPmoc-F₃**. To a freshly-prepared aqueous solution of BPmoc-F₃ and GOx (0.2 mM and 1 mg/mL, respectively, 50 mM HEPES containing 0.5% glycerol, pH 7.4, 110 μL) was added a solution of Zn(NO₃)₂ (8 mM, 10 μL) or H₂O (10 μL). To the resultant mixture was added a solution of glucose (32 mM, 10 μL), and then the resulting solution was incubated at 20 °C for 15, 30, and 60 min. To determine the reaction rate by RP-HPLC, a solution of p-nitrobenzenesulfonamide (0.32 mM, 100 μL in CH₃CN) was added to the reaction solution as an internal standard. The resultant solution was immediately analyzed by RP-HPLC.

**Time-lapse CLSM imaging of the propagating wave**. A freshly-prepared aqueous solution of BPmoc-F₃, BP-TMR, and GOx (1.6 mM, 0.32 μM, and 1 mg/mL, respectively, 50 mM HEPES containing 0.5% glycerol, pH 7.4, 5.5 μL) was deposited between a Matsunami micro cover glasses (bottom: non-coat, 30 × 40 mm, 0.12–0.17 mm, top: 18 × 18 mm, 0.12–0.17 mm, non-coat. The cover glasses were sticked with four double-faced adhesive tape (Nichiban). A mixed solution of Zn(NO₃)₂ and glucose (4 mM and 16 mM, respectively, 1.0 μL in H₂O) was added at the right edge of the droplet one minute after time-lapse imaging was started. The volumes of the droplet containing BPmoc-F₃/BP-TMR/GOx and the solution of Zn(NO₃)₂/glucose were kept constant when applying different concentration of GOx and glucose. The time-lapse movies of a propagating wave were obtained by FV1000. In the case of CLSM imaging with the 100× objective, the images were acquired at ~1 mm from the right edge of the droplet. In the case of the time-lapse movie acquired with the 100× objective, the fluorescent intensity of the ROI [size ($x \times y$): 10 pixels × 512 pixels] at desired $x$ coordinates was calculated by Fiji. When using the 4× objective, fluorescent intensity of the ROI [size ($x \times y$): 10 pixels × 100 pixels] at desired $x$ coordinates was calculated by Fiji.

**CLSM imaging upon treatment of Zn(NO₃)₂ followed by glucose**. The preparation of the droplet was the same as "time-lapse CLSM imaging of the propagating wave". A solution of Zn(NO₃)₂ (8 mM, 0.5 μL in H₂O) was added to the right edge of the droplet. After 30 min, a solution of glucose (32 mM, 0.5 μL in H₂O) was added to the right edge of the droplet one minute after starting time-lapse CLSM imaging. The images were acquired at ~1 mm from the right edge of the droplet by FV1000.

**CLSM imaging upon treatment of glucose followed by Zn(NO₃)₂**. The preparation of the droplet was the same as "time-lapse CLSM imaging of the propagating wave". A solution of glucose (32 mM, 0.5 μL in H₂O) was added to the right edge of the droplet. After 5 min, a solution of Zn(NO₃)₂ (8 mM, 0.5 μL in H₂O) was added to the right edge of the droplet one minute after starting time-lapse CLSM imaging. The images were acquired at ~1 mm from the right edge of the droplet by FV1000.

**Preparation of Oregon Green-modified polystyrene beads**. To a suspension of polystyrene beads (micromod, product code 01-01-502, diameter 500 nm, NH₂-modified, 50 mg/mL, 100 μL) were added 100 mM HEPES buffer (pH 8.0, 100 μL), PEG300-NHS (3.3 mg, 10 μmol, Quanta Biodesign), and Oregon Green 488-NHS (0.3 μmol, 6 μL in DMSO, Thermo Fisher). The mixture was incubated at room temperature for 24 h. The resultant mixture was centrifuged (13500 rpm, 4 °C, 5 min), the supernatant was removed, and then H₂O (1 mL) was added. This washing step was repeated 3 times. The washed beads were collected by centrifuge (13500 rpm, 4 °C, 5 min). To the resulting beads, H₂O (100 μL) was added to obtain a suspension of Oregon Green-modified PEG-coated polystyrene beads.

**In situ CLSM imaging of the bead displacement**. The droplet containing BPmoc-F₃, BP-TMR, GOx, and Oregon Green-modified beads (1.6 mM, 0.32 μM, 1 mg/mL, and 20 μg/mL, respectively, 5.5 μL in 50 mM HEPES, pH 7.4) were prepared by the same protocol as "time-lapse CLSM imaging of the propagating wave." A mixture of Zn(NO₃)₂ and glucose (4 mM and 16 mM, respectively, 1.0 μL in H₂O) was added to the droplet. The images were acquired by LSM800 with a 20× objective lens.

According to Stokes's law at low Reynold number, the force generated by the propagating wave was determined by the following equation.

$$F_d = 6\pi\mu R\nu \tag{5}$$

Here, $F_d$, $\mu$, $R$, and $\nu$ are defined as the force applied to a spherical particle, viscosity of the solution, a radius and velocity of a spherical particle (0.25 μm and 0.4 μm/s), respectively. We determined viscosity of a buffer (50 mM HEPES, pH 7.4), a solution of BPmoc-F₃ before and after Zn(NO₃)₂ to be 1.1, 1.3, 2.5 mPa·s, respectively. Although the exact viscosity of the solution during the propagating

wave could not be determined, we assumed that the maximum viscosity was lower than 2.5 mPa·s because the fluorescent intensity of the nanofibers in the propagating wave was lower than that in the mixture of BPmoc-$F_3$, BP-TMR, and $Zn(NO_3)_2$. Therefore, the force generated by the propagating wave was calculated to be the order of 0.005 pN with the viscosity value of 2.5 mPa·s.

**Determination of the persistence length**. To a solution of BPmoc-$F_3$ and BP-TMR (1.2 mM, 0.34 µM, 50 mM HEPES (pH 7.4), 50 µL) a solution of $Zn(NO_3)_2$ (8 mM, 5.0 µL in $H_2O$) was added. After incubation at room temperature for 30 min, the resultant viscous solution was vortexed for 30 s and transferred on a glass bottom dish. Time-lapse CLSM images were acquired at 33 frames/s by LSM800 with a 100× objective lens. Binary images were skeletonized and further spline interpolated using MATLAB to obtain sub-pixel fiber contour coordinates. The coordinates were used to calculate the 2-dimensional persistence length $L_p$ (Supplementary Fig. 19a):

$$\langle cos[\Delta\theta(s)] \rangle = exp\left(\frac{-s}{2L_p}\right) \tag{6}$$

where $\Delta\theta(s)$ is the angle change over the arc length.

## Data availability
The authors declare that the data supporting the findings of this study are available with the paper and its Supplementary information files. The data that support the findings of this study are available from the corresponding author upon reasonable request.

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

## Acknowledgements

We thank S. Onogi (Kyoto Univ.) and H. Shigemitsu (Osaka Univ.) for their assistance of synthesis of BP-TMR. We thank Andrew Jackson, PhD, from Edanz Group (www.edanzediting.com/ac) for editing a draft of this manuscript. This work was supported by a Grant-in-Aid for Scientific Research on Innovative Areas "Chemistry for Multi-molecular Crowding Biosystems" (JSPS KAKENHI Grant JP17H06348), JST ERATO Grant Number JPMJER1802 to I.H., and by a Grant-in-Aid for Young Scientists (JSPS KAKENHI Grant JP18K14333 and JP20K15400) to R.K.

## Author contributions

I.H. and R.K. designed the project. R.K. conducted the gelation experiments and CLSM imaging. M.M. and M.I. performed numerical simulation and analysis. R.S. and M.T. analyzed the fluorescent beads displacement and the persistence length of supramolecular nanofibers. I.H. and R.K. wrote the manuscript and edited with all authors.

## Competing interests

The authors declare no competing interests.
