## [Peer Review File · Nature Communications]

REVIEWER COMMENTS

Reviewer #1 (Remarks to the Author):

In this manuscript, Hamachi and co-workers present a novel reaction network that can generate force from a transiently formed supramolecular nanofiber's propagation wave. The authors utilized a reaction network comprising of two orthogonal stimuli, coordination triggered (Zn^{2+}) formation and glucose oxidase (GOx)/glucose enzymatic reaction driven decomposition of nanofibers/hydrogel of BPmoc-F3 monomer. With the help of confocal laser scanning microscopic (CLSM) imaging, they could visualize the formation and degradation of supramolecular nanofibers in the presence of minute amount of fluorescent probe (BP-TMR). Next, by a clever experimental design, they observed and quantitatively measured the generation of propagating wave arises from spatially-coupled formation and degradation of supramolecular nanofibers depending on the concentration gradient of chemical stimuli under CLSM. To support this, they carried out a numerical simulation based on a reaction-diffusion model which reinforces the propagative wave dependence on the concentration gradient of the formation and degradation stimuli. Further, they quantitatively measure the force (ca. 0.005 pN), achieved by the propagating wave, by monitoring the displacement of PEG-coated fluorescent labeled polystyrene beads during the wave propagation. The study is novel and extensive with thorough experiments to prove the concept. I recommend the acceptance of this fine study in Nature Communications, after addressing or commenting on the following minor concerns.

Following are some minor comments

1. Supplementary Fig 5c, the intensity at the end point seems to decrease abruptly. What is the reason? What happens if the visualization is done for 1 more hour ?
2. Supplementary Fig 6c, why there is a decreasing intensity in absence of glucose? What happens after another hour? Does the assembly break?
3. Include proper protocol for Figure 2 in supplementary information.
4. What is the reason for the formation of crescent area (right side) in Figure 2e? I assume the addition of stock is dilute or injection of it too forceful to push all material to the left? Otherwise a circular area is expected.
5. How reproducible are the rates of propagating wave?
6. During the different experiments with different concentration of glucose, was the volume of stock injected kept constant?
7. Why the authors account for Zn^{2+} -independent nanofiber elongation process when control experiments disapprove this? Page 6, line 23
8. Fig 4, any attempts were made to change the weight of bead to see the effect on force or change in distance travelled?
9. Is there any way to modulate the force generated from the propagating wave?
10. Is there any effect of concentration of two orthogonal stimuli as well as monomer on the velocity of the wave propagation?

Reviewer #2 (Remarks to the Author):

The manuscript describes transient supramolecular nanofibers that are sustained by two orthogonal reactions- metal coordination and enzymatic oxidation to control assembly and disassembly processes. The chemical concept has been described before, but the novelty here is the generation of propagating waves, some insights into how to achieve this, a kinetic model to describe the results and an analysis of the force generated by these transient nanofibers.

The gel-sol transition is achieved by two chemically orthogonal solicitations taking advantage of two distinct functional groups of the peptide-based monomer: the carboxylic terminus works as a coordination site for the Zn^{2+} ions triggering the hydrogelation, while the generated H_2O_2 upon

treatment with GOx/glucose system leads to the oxidation of the N-terminus and the collapse of the nanofibers. The developed system is based on a previous work by the same group on Nature Chemistry in 2014.

The paper is clearly structured, providing appropriate context to the work presented. The main claims are mostly supported by the data but some aspects need attention, as follows:

1. Is it possible that BP-TMR interferes with measured degradation intensity?
2. Have you explored the possibility of tuning the velocity of the propagating wave (for ex. varying enzyme concentration)?
3. Can authors comment on the possibility of reorienting the migration of the wave?
4. In the abstract, the force is expressed but it is not clear what aspect of the transient system this relates to. Is this force generated by volume, by mass, by fiber?
5. Can they comment on the stochastic nature of these fibers? As they indicate, the van Esch work shows simultaneous growth and reduction of fiber length. Is this also observed in their system?
6. The explanation of the delayed application of the stimuli is not clear in the main text. How much time was left between these two stimuli, and how was this decided? Please explain better the reason for the need to apply both stimuli simultaneously. Presumably the formation reaction (complexation) is instantaneous while the oxidation reaction relies on enzyme diffusion through the formed matrix? Can the comment on difference in diffusion? The focus is on glucose but enzyme diffusion may also play a role?
7. They comment on the need for orthogonality between the two stimuli demonstrated- please provide evidence that these reactions are indeed not impacting on each other.
8. The kinetic model is not fully clear to me. What are the 5 constants? How does the enzymatic conversion rate come into it? Have the k values been determined in isolated experiments. How was the diffusion coefficient of the nanofibers determined?

Reviewers' Comments

Reviewer #1 (Remarks to the Author):

In this manuscript, Hamachi and co-workers present a novel reaction network that can generate force from a transiently formed supramolecular nanofiber's propagation wave. The authors utilized a reaction network comprising of two orthogonal stimuli, coordination triggered (Zn^{2+}) formation and glucose oxidase (GOx)/glucose enzymatic reaction driven decomposition of nanofibers/hydrogel of BPmoc-F3 monomer. With the help of confocal laser scanning microscopic (CLSM) imaging, they could visualize the formation and degradation of supramolecular nanofibers in the presence of minute amount of fluorescent probe (BP-TMR). Next, by a clever experimental design, they observed and quantitatively measured the generation of propagating wave arises from spatially-coupled formation and degradation of supramolecular nanofibers depending on the concentration gradient of chemical stimuli under CLSM. To support this, they carried out a numerical simulation based on a reaction-diffusion model which reinforces the propagative wave dependence on the concentration gradient of the formation and degradation stimuli. Further, they quantitatively measure the force (ca. 0.005 pN), achieved by the propagating wave, by monitoring the displacement of PEG-coated fluorescent labeled polystyrene beads during the wave propagation. The study is novel and extensive with thorough experiments to prove the concept.

I recommend the acceptance of this fine study in Nature Communications, after addressing or commenting on the following minor concerns.

Reply:

Thank you very much for your fruitful comments, which have improved the manuscript. We answered your concerns as shown below.

Comment 1:

Supplementary Fig 5c, the intensity at the end point seems to decrease abruptly. What is the reason? What happens if the visualization is done for 1 more hour ?

Reply:

The decrease of fluorescent intensity at the end time point was probably due to photobleaching of the fluorescent probe (BP-TMR) and/or slight movement along z -axis (perpendicular to the focal plane). We thus conducted the time-lapse imaging again. In

this case, we did not observe any decrease of the fluorescent intensity (supplementary Fig. 5a and 5c). However, a photobleaching slightly occurred when we continued the observation for another one hour (response Fig. 1). We replaced the data of supplementary Fig. 5 and supplementary movie 1 with the new imaging data.

Response Fig. 1. (a) Time-lapse imaging of nanofiber formation upon addition of $\text{Zn}(\text{NO}_3)_2$ for 2 h. (b) Time profile of the total fluorescent intensity of the field of view. Condition: $[\text{BPmoc-F}_3] = 1.6 \text{ mM}$, $[\text{BP-TMR}] = 0.34 \mu\text{M}$, $[\text{Zn}(\text{NO}_3)_2] = 0.8 \text{ mM}$, 50 mM HEPES, pH 7.4, 30 °C.

Comment 2:

Supplementary Fig 6c, why there is a decreasing intensity in absence of glucose? What happens after another hour? Does the assembly break?

Reply:

The decrease of fluorescent intensity in the absence of glucose was due to photobleaching of BP-TMR. According to the reviewer's comment, we conducted the imaging for another 1 h (2.5 h in total), showing that no break of fiber networks was observed (response Fig. 2). To explain the decrease of the fluorescent intensity, we amended the caption of supplementary Fig. 6.

Modification in the supplementary information

Page 15, line 4

The decrease of the fluorescent intensity in supplementary Fig. 6b was due to photobleaching of the fluorescent probe (BP-TMR).

Response Fig. 2. (a) Time-lapse imaging of Zn^{2+} -induced nanofibers after addition of H_2O . (b) Time profile of the total fluorescent intensity of the field of view. Condition: $[\text{BPmoc-F}_3] = 1.6 \text{ mM}$, $[\text{BP-TMR}] = 0.34 \mu\text{M}$, $[\text{GOx}] = 1 \text{ mg/mL}$, $[\text{Zn}(\text{NO}_3)_2] = 0.8 \text{ mM}$, 50 mM HEPES, pH 7.4, 30 °C.

Comment 3:

Include proper protocol for Figure 2 in supplementary information.

Reply:

Thank you for your suggestion. We added the experimental protocol for Figure 2 into the supplementary information. We also modified this protocol to clarify how to calculate the fluorescent intensity.

Modifications in supplementary information

Page 4, line 9 (and page 22, line 31 in the *Method* section)

In the case of the time-lapse movie acquired with the 100× objective, the fluorescent intensity of the ROI [size ($x \times y$): 10 pixels \times 512 pixels] at desired x coordinates was calculated by Fiji. When using the 4× objective, fluorescent intensity of the ROI [size ($x \times y$): 10 pixels \times 100 pixels] at desired x coordinates was calculated by Fiji.

Comment 4:

What is the reason for the formation of crescent area (right side) in Figure 2e? I assume the addition of stock is dilute or injection of it too forceful to push all material to the left? Otherwise a circular area is expected.

Reply:

As mentioned by this reviewer, injection of chemical stimuli pushed BPmoc-F₃, BP-TMR, and GOx to leftward. The shape of the bright crescent region varied (half-circle, crescent, or rectangle) depending on the experiments because injection of the chemical stimuli was conducted by hands; nevertheless the formation of the propagating wave is highly reproducible (please see supplementary Fig. 10 in the revised manuscript). To comment the shape of the crescent area, we modified the caption of Fig. 2 as shown below.

Modification in the main text

Page 18, line 10

The shape of the crescent area varied depending on experiments because injection of the chemical stimuli was conducted manually; nevertheless the formation of the propagating wave is highly reproducible (supplementary Fig. 10).

Comment 5:

How reproducible are the rates of propagating wave?

Reply:

To answer the question, we measured the velocity of the propagating wave another two times. The average velocity was determined to be $54 \pm 8 \mu\text{m}/\text{min}$ (average \pm standard deviation, $n = 3$), indicating the sufficient reproducibility. We modified the main text and added the data to supplementary information (supplementary Fig. 10).

Modification in the main text

Page 6, line 2

On the basis of these observations, the propagating distance and the average velocity of the propagating wave were estimated to be $340 \pm 40 \mu\text{m}$ and $54 \pm 8 \mu\text{m}/\text{min}$, respectively (supplementary Fig. 10).

Comment 6:

During the different experiments with different concentration of glucose, was the volume of stock injected kept constant?

Reply:

Yes. We kept the injection volume constant when applying different concentration of glucose. To clarify this point, we modified the experimental protocol.

Modification in the supplementary information

Page 4, line 5 (and page 22, line 26 in the *Method* section)

The volumes of the droplet containing BPmoc-F₃/BP-TMR/GOx and the solution of Zn(NO₃)₂/glucose were kept constant when applying different concentration of GOx and glucose.

Comment 7:

Why the authors account for Zn²⁺-independent nanofiber elongation process when control experiments disapprove this? Page 6, line 23

Reply:

The term comes from Zn²⁺-independent growth (slow & not evaluated) and from the constant term (approximated zero in the present) resulted in linearization of the Zn²⁺-dependent nonlinear reaction. Under the present propagating-wave conditions, sufficient Zn²⁺ ion made the term negligible small as reviewer's indication. The term was remained for considering 1st order responses for respective factors, which would converge into a single but complex nonlinear reaction term by future works.

Comment 8:

Fig 4, any attempts were made to change the weight of bead to see the effect on force or change in distance travelled?

Reply:

As mentioned in the submitted manuscript, we used fluorescently-labeled polystyrene beads (diameter: 500 nm) to estimate the force by adapting the Stokes law, which describes the frictional force exerted on a particle moving at a constant velocity. Since the force exerted on the surface is given as $F = 6\pi\mu Rv$, to change the "weight" by using other materials with larger densities would not be feasible. For example, the use of silica beads (2.65 gcm⁻³) causes the drift of the bead position in z direction due to sedimentation. In this study, we selected polystyrene with density of 1.06 gcm⁻³ to

minimize the artifact originating from the gravitational force. As the force we estimated is very weak (0.005 pN), we believe the observation of a 500 nm-large polystyrene bead moving at the velocity of 0.4 $\mu\text{m/s}$ is a reasonable choice to calculate the force accurately.

Comment 9:

Is there any way to modulate the force generated from the propagating wave?

Reply:

As mentioned in the main text, the vectorial control, bundling, and branching of nanofibers would modulate the force of the propagating wave. We will challenge such precise control of the nanofiber formation in the future.

Comment 10:

Is there any effect of concentration of two orthogonal stimuli as well as monomer on the velocity of the wave propagation?

Reply:

The experimental window to produce the propagating wave is quite limited (BPMoc-F₃ monomer: 1.6 mM, Zn(NO₃)₂: 0.8 mM, glucose: 3.2 mM). Thus, we cannot modulate the wave velocity by changing concentrations of the monomer and stimuli. Instead, however, we newly found that the decrease of GOx amount slowed the velocity ($20 \pm 2 \mu\text{m/min}$ for 0.5 mg/mL; $54 \pm 8 \mu\text{m/min}$ for 1 mg/mL), suggesting that the fiber degradation is important for the wave velocity (supplementary Fig. 10, 11). We modified the main text as shown below.

Modifications in the main text

Page 6, line 6

The formation of the supramolecular propagating wave was strongly dependent on the reaction conditions. **When half concentration of GOx was used (0.5 mg/mL), the propagating wave was indeed formed, while the duration of the propagating wave slightly increased and the velocity decreased to be $20 \pm 2 \mu\text{m/min}$ (supplementary Fig. 11, supplementary movie 5). On the other hand, a propagating wave was not observed when a smaller amount of glucose (1.0 eq) was used (supplementary Fig. 12a,**

supplementary movie 6).

Reviewer #2 (Remarks to the Author):

The manuscript describes transient supramolecular nanofibers that are sustained by two orthogonal reactions- metal coordination and enzymatic oxidation to control assembly and disassembly processes. The chemical concept has been described before, but the novelty here is the generation of propagating waves, some insights into how to achieve this, a kinetic model to describe the results and an analysis of the force generated by these transient nanofibers.

The gel-sol transition is achieved by two chemically orthogonal solicitations taking advantage of two distinct functional groups of the peptide-based monomer: the carboxylic terminus works as a coordination site for the Zn²⁺ ions triggering the hydrogelation, while the generated H₂O₂ upon treatment with GOx/glucose system leads to the oxidation of the N-terminus and the collapse of the nanofibers. The developed system is based on a previous work by the same group on Nature Chemistry in 2014.

The paper is clearly structured, providing appropriate context to the work presented. The main claims are mostly supported by the data but some aspects need attention, as follows:

Reply:

Thank you very much for your careful reviewing and positive comments. We answered your concerns as follows.

Comment 1:

Is it possible that BP-TMR interferes with measured degradation intensity?

Reply:

Concentration of BP-TMR is much lower than BPmoc-F₃ monomer (BP-TMR: 0.32 μM, BPmoc-F₃: 1.6 mM), thus we expect that BP-TMR would show a negligible impact on the nanofiber degradation. Indeed, we measured the time-lapse imaging of nanofiber degradation with two-fold higher concentration of BP-TMR (concentrations of other molecules are the same). The time-dependence curves were almost identical, indicating that the effect of BP-TMR was minimal (supplementary Fig. 7). We added this time-lapse imaging data into supplementary information.

Comment 2:

Have you explored the possibility of tuning the velocity of the propagating wave (for ex. varying enzyme concentration)?

Reply:

Thank you for your important suggestion. According to your comment, we newly investigated the concentration effect of glucose oxidase (GOx). We found that the wave velocity became slower when decreasing the amount of GOx ($20 \pm 2 \mu\text{m}/\text{min}$ for 0.5 mg/mL; $54 \pm 8 \mu\text{m}/\text{min}$ for 1 mg/mL) (supplementary Fig. 11). In contrast, we could not clearly observe formation of nanofibers when the amount of GOx was higher (2.0 mg/mL) (supplementary Fig. 13). These results indicated that careful modulation of nanofiber degradation kinetics is important for wave formation. We added this result to the main text as shown below.

Modification in the main text

Page 6, line 6

The formation of the supramolecular propagating wave was strongly dependent on the reaction conditions. **When half concentration of GOx was used (0.5 mg/mL), the propagating wave was indeed formed, while the duration of the propagating wave slightly increased and the velocity decreased to be $20 \pm 2 \mu\text{m}/\text{min}$ (supplementary Fig. 11, supplementary movie 5). On the other hand,** a propagating wave was not observed when a smaller amount of glucose (1.0 eq) was used (supplementary Fig. 12a, supplementary movie 6).

Page 6, line 16

When a larger amount of **GOx (2 mg/mL)** or glucose (63 eq) was used, we observed negligible formation of nanofibers owing to rapid degradation of BPmoc-F₃ (supplementary Figs. 13, 14).

Comment 3:

Can authors comment on the possibility of reorienting the migration of the wave?

Reply:

We thank the reviewer for an interesting suggestion. With our current system, it will be difficult to "change" the propagating direction. On the other hand, since the wave propagation is driven by the reaction-diffusion mechanism (Figure 3), it may be

expected that the small perturbation of diffusion (by microfluidics, for example) would enable the modulation of wave propagation. We mention about this possibility in the revised manuscript.

Modification in the main text

Page 8, line 2

Furthermore, the much smaller diffusion coefficient of the nanofibers is another important factor for producing a propagating wave, suggesting that a supramolecular nanofiber is one of the most suitable scaffolds for spatiotemporal pattern formation. **The numerical simulation also suggested that the small perturbation of diffusion (by microfluidics, for example) would enable the modulation of wave propagation.** Our numerical simulation demonstrates that the design principle of the reaction network based on orthogonal formation and degradation stimuli is a promising strategy for forming supramolecular propagating waves.

Comment 4:

In the abstract, the force is expressed but it is not clear what aspect of the transient system this relates to. Is this force generated by volume, by mass, by fiber?

Reply:

Thank you for your suggestion, According to the reviewer's comment, we modified abstract as shown below.

Modification in the main text

Page 2, line 11

Moreover, the force (0.005 pN) generated by **chemophoresis and/or depletion force of** this propagating wave can move nanobeads along the wave direction.

Comment 5:

Can they comment on the stochastic nature of these fibers? As they indicate, the van Esch work shows simultaneous growth and reduction of fiber length. Is this also observed in their system?

Reply:

We appreciate your useful comment. As described by the reviewer, van Esch demonstrated the simultaneous growth and shrinkage of nanofibers in proximity. In our propagating wave, the nanofiber formation and degradation were highly regulated in a spatiotemporal manner, thus such a stochastic behavior did not occur. We consider that the steep concentration gradient of chemical stimuli would suppress the stochastic nature of nanofiber formation/degradation processes. We added these comments in the main text.

Modification in the main text

Page 5, line 17

In contrast to simultaneous growth and shrinkage of nanofibers in proximity reported by van Esch,²² nanofiber formation and degradation in this propagating wave were highly regulated in a spatiotemporal manner by the concentration gradient of stimuli, which would suppress the stochastic nature of the fiber formation/degradation process.

Comment 6:

The explanation of the delayed application of the stimuli is not clear in the main text. How much time was left between these two stimuli, and how was this decided? Please explain better the reason for the need to apply both stimuli simultaneously. Presumably the formation reaction (complexation) is instantaneous while the oxidation reaction relies on enzyme diffusion through the formed matrix? Can the comment on difference in diffusion? The focus is on glucose but enzyme diffusion may also play a role?

Reply:

We set the delay time to be 30 min because nanofiber formation is almost saturated (supplementary Fig. 5). Our results suggested that the concentration gradient of not only the degradation stimulus but also the formation stimulus ($\text{Zn}(\text{NO}_3)_2$) may be important for the wave formation. We thus expect that a propagating wave should be formed if the time difference between two stimuli is small. However, it is experimentally difficult to inject two stimuli with a short time lag. Therefore, we adopted the simultaneous addition of two stimuli in this manuscript. As pointed out by the reviewer, the concentration gradient of GOx may form after addition of stimuli so that it is possible that GOx diffusion may play a role for wave formation. This point is very interesting, thus we will try to assess the effect of GOx diffusion in the near future.

Modification in the main text

Page 6, line 18

To confirm whether simultaneous addition of $\text{Zn}(\text{NO}_3)_2$ and glucose is **important** for the wave formation, we treated the droplets of BPmoc-F₃, BP-TMR, and GOx by addition of glucose **30 min after $\text{Zn}(\text{NO}_3)_2$ injection (nanofiber formation is saturated after incubation for 30 min as shown in supplementary Fig. 5)**. The BPmoc-F₃ nanofibers formed upon treatment with $\text{Zn}(\text{NO}_3)_2$ and begun to homogenously disappear 50 min after addition of glucose, indicating no generation of a propagating wave (Fig. 2d, supplementary Fig. 15, supplementary movie 7). **It implies that concentration gradient of $\text{Zn}(\text{NO}_3)_2$ would play an important role for wave formation.**

Comment 7:

They comment on the need for orthogonality between the two stimuli demonstrated- please provide evidence that these reactions are indeed not impacting on each other.

Reply:

To demonstrate the orthogonality of the stimuli, we measured (1) formation kinetics of Zn^{2+} -induced nanofibers in the presence and absence of glucose or GOx by time-lapse CLSM imaging and (2) degradation kinetics of BPmoc-F₃ by GOx/glucose in the presence/absence of Zn^{2+} ion by RP-HPLC. In both experiments, the kinetics of nanofiber formation and BPmoc-F₃ degradation hardly changed even in the presence of the other type of stimulus, suggesting that the interaction between the two chemical stimuli is negligible (supplementary Fig. 8 and 9). We added the comments into the main text and the data into supplementary information.

Modification in the main text

Page 4, line 31

We also confirmed that the interaction between $\text{Zn}(\text{NO}_3)_2$ and the GOx/glucose pair was negligible in nanofiber formation and degradation processes (supplementary Figs. 8, 9).

Comment 8:

The kinetic model is not fully clear to me. What are the 5 constants? How does the enzymatic conversion rate come into it? Have the k values been determined in isolated

experiments. How was the diffusion coefficient of the nanofibers determined?

Reply:

In our numerical simulation, the reaction system is simplified. For the degradation reaction, the enzymatic reaction is not visibly involved and instead, the simple degradation processes of monomer and fibers are considered. We therefore assumed the five rate constants which correspond to 1st order responses on the respective factors for the reaction speeds. We did not use any measured values for the rate constants in this reaction-diffusion equation. Indeed, it is extremely difficult to experimentally determine the kinetics and the nonlinearity of supramolecular nanofiber formation. We approximated the fiber formation process as the first order against the monomer concentration because the monomer concentration is limited under the initial concentration. Based on our time-lapse imaging, the diffusion constant of supramolecular nanofibers can be considered to be almost zero on the experimental timescale, and thus we assumed that the diffusion constant of nanofibers is much smaller than those of other chemical species in the simulation. Given that the scale and diffusion shape of the propagation wave obtained by the simulation shows the close approximations of the experiment, our numerical simulation supports that our design principle of the reaction network is promising for generation of supramolecular propagating waves.

REVIEWERS' COMMENTS:

Reviewer #1 (Remarks to the Author):

I really appreciate the efforts taken by the author to address all the comments. All my comments are addressed in a satisfactory manner in the revised submission.

I strongly recommend the acceptance of this fine study in Nature communications.

Reviewer #2 (Remarks to the Author):

The authors have adequately addressed my comments and I support publication in Nature Communications